# Dynamic Response Analysis of an Immersed Electrothermally Actuated MEMS Mirror

**Tailong Liu** [1], **Teng Pan** [2], **Shuijie Qin** [3,*], **Hui Zhao** [2,4,5] **and Huikai Xie** [2,4,*]

1  School of Mechanical Engineering, Guizhou University, Guiyang 550001, China
2  School of Integrated Circuits and Electronics, Beijing Institute of Technology, Beijing 100811, China
3  Guizhou Province Key Laboratory for Photoelectric Technology and Application, Guizhou University, Guiyang 550025, China
4  BIT Chongqing Center for Microelectronics and Microsystems, Chongqing 400030, China
5  LightVision Technologies, Inc., Foshan 528000, China
*  Correspondence: shuijie_qin@sina.com (S.Q.); hk.xie@ieee.org (H.X.)

**Abstract:** MEMS mirrors have a wide range of applications, most of which require large field-of-view (FOV). Immersing MEMS mirrors in liquid is an effective way to improve the FOV. However, the increased viscosity, convective heat transfer and thermal conductivity in liquid greatly affect the dynamic behaviors of electrothermally actuated micromirrors. In this paper, the complex interactions among the multiple energy domains, including electrical, thermal, mechanical and fluidic, are studied in an immersed electrothermally actuated MEMS mirror. A damping model of the immersed MEMS mirror is built and dimensional analysis is applied to reduce the number of variables and thus significantly simplify the model. The solution of the fluid damping model is solved by using regression analysis. The dynamic response of the MEMS mirror can be calculated easily by using the damping model. The experimental results verify the effectiveness and accuracy of these models. The difference between the model prediction and the measurement is within 4%. The FOV scanned in a liquid is also increased by a factor of 1.6. The model developed in this work can be applied to study the dynamic behaviors of various immersed MEMS actuators.

**Keywords:** MEMS mirror; micromirror; electrothermal actuation; immersion; MEMS in liquid

## 1. Introduction

The advent of microelectromechanical system (MEMS) spatial light modulators, such as scanning micromirrors [1] and digital micromirror devices (DMDs) [2], has allowed a wide range of fast and accurate optical modulation applications, such as coded apertures [3], analysis of fast-moving scenes [4], and compressive sensing [5]. These optical MEMS devices provide an excellent platform to extend active computer vision techniques to the low-power and miniature domains [6]. MEMS mirrors can be electrostatically, piezoelectrically, electromagnetically or electrothermally actuated. Among them, electrostatic MEMS mirrors typically have a small fill factor and need high driving voltage [7]. Electromagnetic MEMS mirrors usually have a large size because of the need for permanent magnets [8]. Most piezoelectric MEMS mirrors either have small mirror size or small scan angle [9]. On the other hand, electrothermally actuated MEMS mirrors have a high fill factor and large scanning angle with a low driving voltage [10,11]. A field of view (FOV) over 60° is often needed for these MEMS-enabled low-power vision platforms, but most MEMS micromirrors have limited FOV in the order of only 10°. Thus, enlarging the FOV of a MEMS micromirror is critical for many applications.

Zhang et al. proposed placing an electrothermally actuated bimorph MEMS mirror in water and improved the FOV by a factor over 2 [12]. However, putting an electrothermal MEMS mirror in a liquid has a significant impact on the MEMS mirror's dynamic response, which may lead to unexpected scanning behaviors. M. Li et al. studied the thermal and

mechanical response of an electrothermally actuated bimorph MEMS mirror working in air [13], but the viscosity, density and convective heat transfer of a liquid are all much larger than those in air. In addition, the temperature of the thermal bimorph actuator changes in a large range, leading to a non-isothermal flow, while the viscosity of the liquid is influenced by the heating source of the thermal bimorph actuator. These complex electro-thermo-mechanical-fluidic interactions make the prediction of the dynamic characteristics of such electrothermal MEMS micromirrors working in liquid difficult. Thus, a prediction model is needed to evaluate the working characteristics of electrothermal MEMS mirrors in various liquids.

Thermal dynamic response can be easily represented as an equivalent circuit model, including the heat capacitance of the bimorph actuator and the thermal resistance from the bimorph to the substrate and the ambient [14]. In the mechanical domain, the equivalent circuit modeling method can be applied as well, where the primary effort is to determine the equivalent damping resistance of the MEMS device. Squeeze-film damping effects on the dynamic responses of electrostatic MEMS mirrors and the damping of the electromagnetically MEMS mirrors have been studied extensively [15–22], but this does not apply to electrothermally actuated MEMS mirrors as their actuation beams are largely separated from the surrounding microstructures. Additionally, the Reynolds numbers in electrothermally actuated MEMS mirrors are quite different from those in electrostatic MEMS mirrors. Moreover, upon working in liquid, the joule heating of the thermal bimorph actuator will generate considerable temperature change and temperature gradient, which further alter the viscosity of the liquid.

In this paper, an analytical model of the fluidic damping of the electrothermal micromirror operating in liquid is developed. This model, applied with the electro-thermo-mechanical model of the electrothermal micromirror, can readily predict the dynamic behaviors of the immersed micromirror. The accuracy of the model is validated experimentally. According to the model, the relation between the effective damping coefficient and the properties of the liquid can be obtained easily. Additionally, the model is applicable to nearly all MEMS actuators immersed in a Newtonian fluid.

This paper is organized as follows. The analytical models of the electrothermally actuated MEMS mirror are introduced in Section 2. The damping model of the immersed micromirror is built and a numerical solution method is proposed by using dimensional and regression analysis in Section 3. The accuracy of the prediction model is verified in Section 4.

## 2. The 2D Electrothermally Actuated MEMS Mirror

The two-axis electrothermally actuated MEMS mirror is composed of a square mirror plate and four driving arms, as illustrated in Figure 1a. The driving arms consist of the inverted-series-connected (ISC) thermal bimorph actuator structure [1], labelled as ACT1, ACT2, ACT3 and ACT4 as shown in Figure 1a. The mirror plate is connected to the substrate symmetrically by the actuators. The ISC bimorph is composed of three parts: an inverted (IV) segment, a non-inverted (NI) segment and an overlap (OL) segment, and the shape of the bimorph is an "S", as shown in Figure 1b. Figure 1b also shows the materials of the bimorph, which include aluminum (Al) and silicon dioxide ($SiO_2$), and the large thermal expansion coefficient (TEC) difference between these two materials. The embedded heater of the bimorph is the titanium (Ti). Figure 1c is a scanning electron microscopy (SEM) image of a fabricated MEMS mirror. The side length, $L$, of the AL-coated mirror plate is 500 μm; the length of the bimorph, $L_b$, is 258 μm; and the elevation of the mirror plate is 235 μm, caused by the residual stresses produced during the device fabrication. Table 1 is the structure parameters of the micromirror.

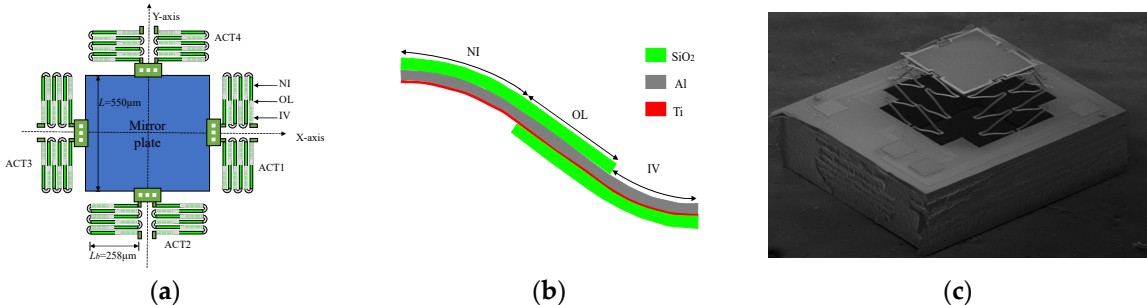

**Figure 1.** (**a**) The structure of the two-axis electrothermal MEMS mirror. (**b**) One "S" shaped inverted-series-connected (ISC) bimorph design. (**c**) An SEM of the MEMS mirror.

**Table 1.** The structure parameters of the micromirror.

| Structure | Mirror Plate | Bimorph | Total Size |
|---|---|---|---|
| Size (length × width × height ($\mu m^3$)) | $550 \times 550 \times 20.1$ | $245 \times 20 \times 3$ | $700 \times 700 \times 400$ |

The fabrication process of the MEMS mirror combines the surface- and bulk-micromachining processes, which is described in detail in [1]. The mirror plate is coated with a highly reflective Al layer and supported by a single-crystal silicon layer. The residual stresses, which determine the initial elevation of the mirror plate, are produced in the fabrication steps [23].

A resistive heater is embedded in each actuator. Joule heating is generated when a voltage is applied to the resistor, causing the temperature of the actuator to increase. Then, the thermal bimorph actuator structure will bend due to the TEC difference between the two bimorph materials, leading to the movement of the mirror plate. The bending angle is dependent on the bimorph materials, structure design and dimensions of the actuators [23].

According to the principle of the electrothermally actuated micromirror, when a voltage is applied to one of the actuators, e.g., ACT1, a thermal-induced force, $F_t$, is generated and ACT1 moves downward as shown in Figure 2a, and the mirror plate is deflected by an angle, $\theta$, with an angular velocity, $\omega$, causing the two neighboring actuators, ACT2 and ACT4, to displace by half of that of ACT1, as shown in Figure 2b. Meanwhile, a damping force, $F_D$, is resultant from the moving mirror plate and the actuators. In general, the actuator has some lateral shift in $r$ directions. However, for this structure, the lateral shift can be negligible. In addition, the thickness of the mirror plate is significantly larger than the actuators so that the mirror plate can be regarded as a rigid body. In this case, the edge of the mirror plate connected with ACT3 is the rotational axis, and the distance of a point on the mirror plate from the rotational axis is denoted as $r$. The velocity of the point on the mirror plate can be expressed as $v = r \cdot \omega$ and the velocities of the points connected with ACT1, ACT2, and ACT4 are $v_{ACT1} = L \cdot \omega$ and $v_{ACT2} = v_{ACT4} = \frac{1}{2} L \cdot \omega$, respectively.

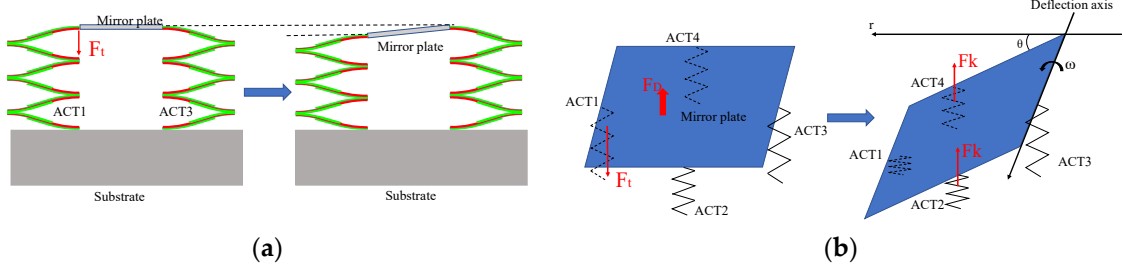

**Figure 2.** (**a**) The side view of the S-shaped actuation structure; (**b**) the schematic of the working electrothermally actuated micromirror.

Figure 3a shows the measured frequency response of the MEMS mirror in air with a sine waveform voltage at 0–5 V from 1 Hz to 3000 Hz, and there are two resonant peaks in this range. The first is 1186 Hz and the second is 2370 Hz. The Q factors are about 50 [13]. Therefore, the system is underdamped in air. The 3-dB cut-off frequency is 100 Hz. This low cut-off frequency is due to the relatively large thermal response time of the electrothermal bimorph actuators.

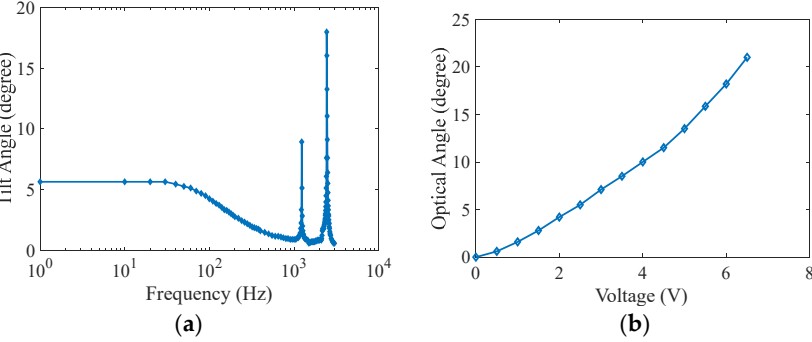

(a)　　(b)

**Figure 3.** Frequency response (**a**) and static response (**b**) of the optical angular scan of the electrothermally actuated MEMS mirror.

Figure 3b shows the quasi-static response of the MEMS mirror. The maximum static optical scan angle by one-side actuation can reach 20° at only 6.5 V, or a FOV of 40° with actuation from both sides. It can be seen that this maximum scan angle is well below the desired 60° FOV needed for many applications. In this work, it is proposed that the FOV can be magnified by immersing the MEMS mirror inside a liquid whose refractive index is greater than that of air.

Figure 4 shows the principle of Snell's law. When the mechanical angle of the mirror plate in the liquid is $\theta_M$, the output optical angle is $\arcsin[n \cdot \sin 2\theta_M]$, where $n$ is the refractive index of the liquid. Mineral oils are usually used as the immersed liquids due to their transparency, moderate refractive index values and relatively low viscosity. Table 2 lists the properties of several frequently used mineral oils [12]. For the MEMS mirror described above, when the refractive index of the liquid is 1.5, the maximum FOV in liquid can reach 60°.

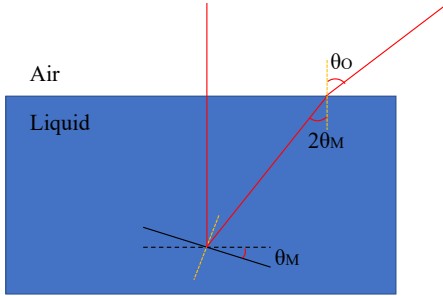

**Figure 4.** Principle of Snell's law.

**Table 2.** Physical properties of several mineral oils.

| Name | Refractive Index | Viscosity (kg/(m·s) at 293K) | >Density (kg/ m³) | Heat Transfer Coefficient (W/(m·K)) |
|---|---|---|---|---|
| Dimethyl silicone oil (1000 cs) | 1.5 | 0.94 | 940 | 0.18 |
| Dimethyl silicone oil (5 cs) | 1.39 | 0.0047 | 940 | 0.182 |
| White oil | 1.45 | 0.00343 | 816 | 0.16 |
| Liquid paraffin | 1.4 | 0.0188 | 847 | 0.185 |

According to the working principle of the electrothermally actuated MEMS mirror, the transfer function of the micromirror system can be simplified as [13]:

$$H(s) = H_T(s) \cdot H_M(s) = \frac{(R_T + R_{ba})}{(R_T + R_{ba})C_T s + 1} \cdot \frac{L/I}{s^2 + \left(\frac{D}{I}\right)s + k_\theta/I} \tag{1}$$

where $H_T(s)$ and $H_M(s)$ are the transfer functions of the thermal response and mechanical response of the system, respectively, $C_T$ is the heat capacitance of the bimorph actuator, $R_T$ is the equivalent thermal resistance of the structure of the micromirror, $R_{ba}$ is the equivalent thermal resistance from the bimorph to the ambient, $I$ is the moment of inertia of the square mirror plate, $D$ is the damping coefficient and $k_\theta$ is the equivalent torsional stiffness of all bimorph actuators.

From Equation (1), it can be seen that the thermal response is affected by $R_T$, $R_{ba}$, and $C_T$. The mechanical response will change with $L$, $D$, $k_\theta$ and $I$. When the working environment is changed, $R_{ba}$ and $D$ are changed accordingly. Other variables in Equation (1) are unchanged when a particular micromirror is selected and can be calculated according to [15].

$R_{ba}$ is simply given by the following equation [24]:

$$R_{ba} = \frac{1}{hs} \tag{2}$$

where $h$ is the thermal conductivity coefficient of the fluid because the convective heat transfer between the bimorph and fluid is negligible compared to the conductive heat transfer [24]. Thus, as long as the fluid is selected, the thermal response can be readily determined. In contrast, the mechanical response is more complicated to model and calculate as the determination of the damping coefficient $D$ is affected by the couplings among the thermal, mechanical and fluidic response, which is the focus of the next section.

### 3. Analyses of the Damping Coefficient

Immersing a MEMS mirror improves its scanning FOV, but the change of the working environment affects the damping of the working micromirror. So the mechanical response of the micromirror will change. Note that the scanning micromirror in air is an underdamped system while it becomes an overdamped system in liquid. Thus, the damping due to the liquid dominates and other damping factors are ignored in this work. The damping model will be established next.

To model the viscous liquid damping, the flow regime in a microsystem must be determined first. The Reynolds number, $Re$, of a liquid is given by:

$$Re = \frac{\rho v_{avg} L}{\mu} \tag{3}$$

where $\mu$ is the fluid viscosity, $L$ is the characteristic length, $v_{avg}$ is the average velocity and $\rho$ is the fluid density. For the scanning micromirror, the Reynold's number $Re$ is given by:

$$Re = \frac{2\rho\hat{\theta}fL \cdot L}{\mu} \tag{4}$$

where $\hat{\theta}$ is the amplitude of the deflection angle, $f$ is the working frequency and $2\hat{\theta}fL$ is the average velocity in Equation (3). Assume the MEMS mirror works at 100 Hz with a deflection angle of $\pm 10°$ in the liquid paraffin (1 atm, 25° C), $\hat{\theta} = 10°$ or 0.17 rad and $L = 0.5 \times 10^{-3}$ m. Therefore, $Re$ is about 0.36. Thus, it is a creeping flow for this scanning micromirror [25].

Meanwhile, the Knudsen number, $K_n$, is given by:

$$K_n = \frac{\lambda}{l} \tag{5}$$

which is the ratio of the mean free path $\lambda$ of the molecules to the channel length, $l$. For the mineral oils listed in Table 2, $K_n$ is about $1.2 \times 10^{-5}$, which is far less than $10^{-3}$, so the fluid flow is continuous.

According the above calculation, the assumptions are proposed as follows:

(1)    The inertial terms of the damping are neglected due to the small *Re* [26]; and
(2)    The liquid is incompressible and continuous.

The damping of the scanning mirror system can be divided into two situations: isothermal flow damping and non-isothermal flow damping. An isothermal flow is formed around the mirror plate and the bimorph actuators when the bimorph actuators are not actuated. A non-isothermal flow is generated when the actuators are applied with voltage. The damping of a continuous creeping flow can be calculated as per [27], but the analytical solution can be obtained only under specific boundary conditions. The electrothermally actuated MEMS mirror has complicated boundaries. In addition, since the electrothermally actuated MEMS mirror changes the temperature of the surrounding fluid, the temperature dependances of the thermal conductivity and viscosity need to be reconsidered. These complex situations make analytical solutions difficult to obtain. The numerical solution may be obtained by the finite difference method, but it is difficult to determine the influence of each variable on the damping. Additionally, the accuracy of the result is affected by the approximated boundary conditions, especially when the shape of the actuator is irregular. To overcome these shortcomings, a dimensional analysis followed by a regression analysis is employed.

### 3.1. The Damping of the Isothermal Flow

The isothermal flow is considered first. The variables affecting the damping in the isothermal creeping flow include the velocity $v$, the characteristic length $L$ for mirror plate (or $L_b$ for bimorphs) and the fluid viscosity $\mu$ [27]. The dimensional analysis according to Buckingham's $\pi$-Theorem is used, which can reduce the number of variables to attain the relationship of the variables [25]. Table 3 shows the variables and their dimensions.

**Table 3.** Dimensions of the variables.

| Variable | $F_D$ | $v$ | $L$ | $\mu$ |
|----------|-------|-----|-----|-------|
| Dimensions | $\frac{mL}{t^2}$ | $\frac{L}{t}$ | $L$ | $\frac{m}{Lt}$ |

There are three basic dimensions listed in Table 3, i.e., $m$ is mass, $L$ is length and $t$ is time. One dimensionless parameter, $\Pi_1$, is defined as follows:

$$\Pi_1 = F_D v^a L^b \mu^c \tag{6}$$

and its dimensions can be written as $\left\{ \frac{mL}{t^2} \frac{L^a}{t^a} L^b \frac{m^c}{L^c t^c} \right\}$. Let the summed index of each basic dimension be zero, leading to $a = b = c = -1$. Thus, Equation (6) becomes:

$$\Pi_1 = \frac{F_D}{v \mu L} \tag{7}$$

According to Buckingham's $\pi$-Theorem [25], $\Pi_1$ is a constant when there is only one dimensionless parameter. Then, we can rewrite Equation (7) as:

$$F_D = c \cdot v \mu L \tag{8}$$

where $c$ is a constant which is only related to the geometric shapes of the moving plate [25]. Based on Equation (8), the damping force of the scanning mirror plate is given by:

$$dF_{D\_m} = -c_m \cdot v_m \cdot \mu \cdot dr = -c_m \cdot \mu_0 \cdot r \cdot \omega \cdot dr \tag{9}$$

where $c_m$ is the constant to solve the damping of the scanning mirror and $v_m = r \cdot \omega$ is the velocity of a point on the mirror plate at distance $r$. Thus, the damping torque of the scanning mirror is given by:

$$M_{D_m}(\omega) = -\left| \int_0^L r \cdot dF_D \right| = -\frac{1}{3} c_m \cdot \mu_0 \cdot L^3 \cdot \omega \tag{10}$$

The damping torque of the two neighboring actuators ACT2 and ACT4 can be expressed as:

$$M_{D_{2,4}}(\omega) = -2 \cdot F_{D\_2,4} \cdot \frac{L}{2} = -\frac{1}{2} c_b \cdot \mu_0 \cdot L_b \cdot L^2 \cdot \omega \tag{11}$$

where $c_b$ is the constant to solve the damping of the bimorphs.

### 3.2. The Damping of the Non-Isothermal Flow

When calculating the damping of the working actuator (ACT1), the change of the viscosity with the temperature must be considered, and the relation between the viscosity and the temperature, $T$, can be numerically fitted as [28]:

$$\mu = b(T - T_0)^\varphi$$

where $b$ is a constant only related to the reference viscosity, $\mu_0$, at the ambient temperature, $T_0$, and $\varphi$ is an index as a measure of the change in viscosity with temperature. Furthermore, the viscosities of the liquid are different at different positions because of the thermal conduction, as shown in Figure 5. A segment of the bimorph is chosen for describing the non-isothermal flow. The temperature gradient is generated due to the thermal conductance when there is a temperature difference between the working bimorph and the environment. There also exists a viscosity gradient around the bimorph. Therefore, in this work, the effective viscosity, $\mu_e$, was introduced for the non-isothermal flow by regression analysis. In this way, the complex calculation of the gradient field can be avoided. The effective viscosity is dependent on $\mu_0$, $\varphi$, $h$ and the temperature difference from the reference temperature, $T_d$.

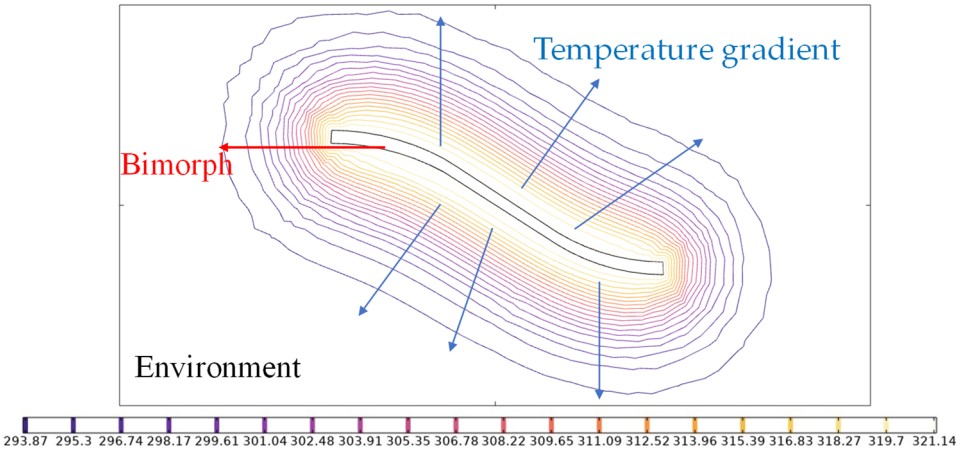

293.87 295.3 296.74 298.17 299.61 301.04 302.48 303.91 305.35 306.78 308.22 309.65 311.09 312.52 313.96 315.39 316.83 318.27 319.7 321.14

**Figure 5.** The isotherm around the working bimorph.

The damping torque of the working actuator (ACT1) is given by:

$$M_{D_1}(\omega) = -F_{D\_1} \cdot L = -c_b \cdot \mu_e \cdot L_b \cdot L^2 \cdot \omega \tag{12}$$

where $\mu_e$ is expressed as:

$$\mu_e = f(\mu_0,\ T_d,\ h,\ \varphi) \tag{13}$$

The actual dependence of $\mu_e$ on the variables in Equation (13) can be found using regression analysis. In the following analysis, the coefficient of determination, $R^2$, was set at 0.99 so that the regression analysis had a high accuracy. First, the nonlinear relationship between the independent variable with each dependent variable was analyzed. A 3D model of the MEMS mirror was established in COMSOL. The dimensions and the materials were the same as those of the MEMS mirror described in Section 2. The fluid was liquid paraffin. The simulation was analyzed using Fluid-Structure, Non-isothermal Flow, and Thermal Expansion Interaction in the COMSOL, in which the fluid field, the solid mechanics field and the heat transfer in solids and fluids were all considered. The fluid is an incompressible creeping flow according to its *Re* number. To simulate the motion of the fluid more accurately, a moving mesh was used for the fluid which can automatically update its size. The temperature of the actuators was set at 470 K and the environment temperature at 293.15 K. Performing the simulation yielded the values of the constants, which were $c_m = 8.42$ and $c_b = 1.23$. The CFD simulation also generates the results of the changes of $\mu_e$ with respect to $\mu_0$, $T_d$, $h$ and $\varphi$, respectively, which are shown in Figure 6.

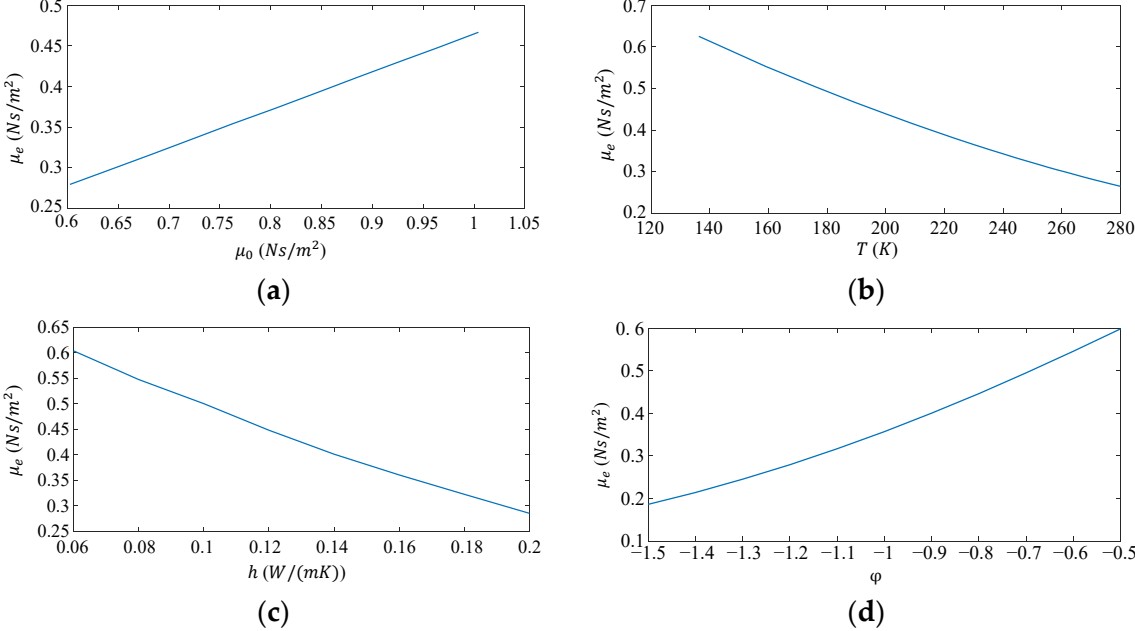

**Figure 6.** The simulation result of changing $\mu_e$ with (**a**) $\mu_0$, (**b**) $T_d$, (**c**) $h$ and (**d**) $\varphi$.

According to the above data, the fitting functions can be obtained by using the MATLAB Curve Fitting Toolbox as follows:

$$\begin{cases} \mu_e = 0.4652 \cdot \mu_0 - 0.001 \\ \mu_e = -1.856 \cdot T_d^{0.1344} + 4.219 \\ \mu_e = -1.634 \cdot h^{0.5952} + 0.9 \\ \mu_e = 3.349 e^{1.641\varphi} - 2.624 \times e^{2.193\varphi} \end{cases} \tag{14}$$

After obtaining the relationship between $\mu_e$ and each independent variable, we adopt the equation with the following form to express $\mu_e$:

$$\mu_e = C_1 + C_2 \cdot \mu_0 + C_3 \cdot T_d^{0.1344} + C_4 \cdot h^{0.5952} + C_5 \cdot e^{1.641\varphi} + C_6 \cdot e^{2.193\varphi} \tag{15}$$

where $C_1$, $C_2$, $C_3$, $C_4$, $C_5$: and $C_6$ are the coefficients. Then, multiple linear regressions can be used and the following fitting function is obtained:

$$\mu_e(\mu_0,\ T_d,\ h,\ \varphi) = 4.026 + 0.4824{\cdot}\mu_0 - 1.83{\cdot}T_d{}^{0.1344} - 1.6725{\cdot}h^{0.5952} + 3.383{\cdot}e^{1.641\varphi} - 2.6645{\cdot}e^{2.193\varphi} \tag{16}$$

According to Equations (10)–(12), the total damping torque is given by:

$$
\begin{aligned}
M_{D_{total}}(\mu_0,\ \mu_e,\ L_b,\ L,\ \omega) \quad &= M_{D\_m} + M_{D\_2,4} + M_{D\_1} \\
&= -\left(\tfrac{1}{3}c_m{\cdot}\mu_0{\cdot}L^3 + \tfrac{1}{2}c_b{\cdot}\mu_0{\cdot}L_b{\cdot}L^2 + c_b{\cdot}\mu_e{\cdot}L_b{\cdot}L^2\right){\cdot}\omega
\end{aligned}
\tag{17}
$$

and finally the damping coefficient is obtained as:

$$D(\mu_0,\ \mu_e,\ L_b,\ L) = \frac{M_{D_{total}}}{\omega} = -\left(\frac{1}{3}c_m{\cdot}\mu_0{\cdot}L^3 + \frac{1}{2}c_b{\cdot}\mu_0{\cdot}L_b{\cdot}L^2 + c_b{\cdot}\mu_e{\cdot}L_b{\cdot}L^2\right) \tag{18}$$

According to Equation (18), it can be clearly seen that the fluid damping coefficient changes with the working environment. Then, substituting Equation (18) into Equation (1), the step response of the scanning micromirror in liquid is obtained as:

$$\widetilde{\theta}(t) = 1 - \frac{\tau\omega_n e^{-\frac{t}{\tau}}}{\frac{1}{\tau\omega_n} - 2\zeta + \tau\omega_n} - \frac{\frac{e^{-\zeta\omega_n t}}{\tau\omega_n}}{\sqrt{1-\zeta^2}\sqrt{\frac{\frac{1}{\tau\omega_n}-2\zeta}{\tau\omega_n}+1}}\sin\left(\omega_n\sqrt{1-\zeta^2}t+\alpha\right) \tag{19}$$

where $\tau = C_T(R_{bs} + R_{ba})$, $\omega_n = \sqrt{k_\theta/I}$ and $\zeta = \frac{D}{2\sqrt{k_\theta/I}}$ are the thermal time constant, natural resonant frequency and damping ratio of the electrothermally micromirror system, respectively. The quality factor is given by $Q = \frac{1}{2\zeta}$.

## 4. Experimental Verification

To validate the accuracy of the model for predicting the dynamic response, an experiment was designed and implemented. The numerical damping model discussed in Section 2 was applied to calculate the damping coefficient $D$ to be used in Equation (1), which represents the dynamic model of the electrothermally actuated micromirror. The MEMS mirror in this experiment is described in Section 2. As shown in Figure 7a, the MEMS mirror was immersed in dimethyl silicone oil (1000 cs) and a position-sensitive device (PSD, ON-TRAK OT301DL) was employed to detect the scanning angle, as shown in Figure 7b. Dimethyl silicone oil has a relatively large viscosity and it clearly changes with the temperature. In this case, $\varphi = -1.193$. Thus, the non-isothermal flow applies. For this situation, the damping coefficient changed with the temperature of the working bimorphs.

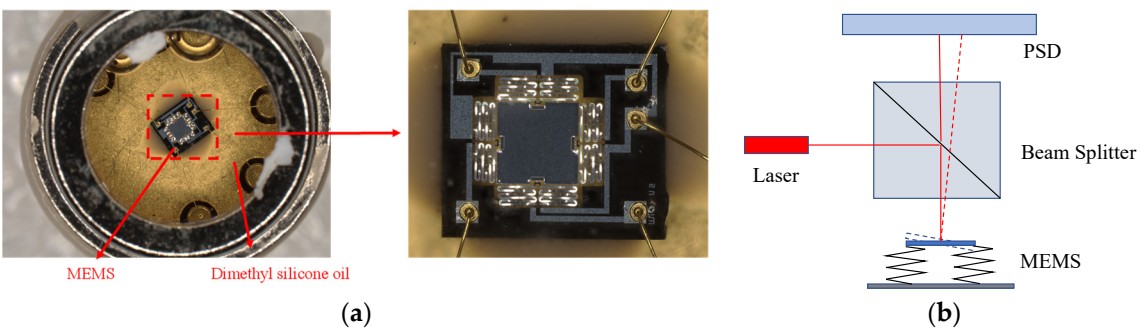

**Figure 7.** (**a**) The MEMS mirror immersed in dimethyl silicone oil; (**b**) the schematic of the PSD experimental platform.

A sine waveform voltage (amplitude: 4.75 V; dc offset: 4.75 V) with a frequency sweeping from 0.1 Hz to 3000 Hz was applied to the MEMS mirror. Then, the tilt angle of the MEMS mirror was measured using the PSD and the frequency response was obtained [29]. Meanwhile, the same voltage signal was used as the input to the dynamic model and the

predicted frequency response was obtained. Figure 8 shows the frequency response and quasi-static response. No resonant peaks were observed as shown in Figure 8a because of the large damping [13]. The cutoff frequency was 4 Hz. The FOV in the liquid could reach as high as 64°, compared to 40° in air, as shown in Figure 8b, which increased by a factor of 1.6. According to Equation (2), the equivalent thermal resistance from the bimorph to the ambient of the micromirror is small because of the relatively large thermal conductivity of the liquid. Therefore, higher voltage can be applied to the bimorph actuator, as shown in Figure 8b.

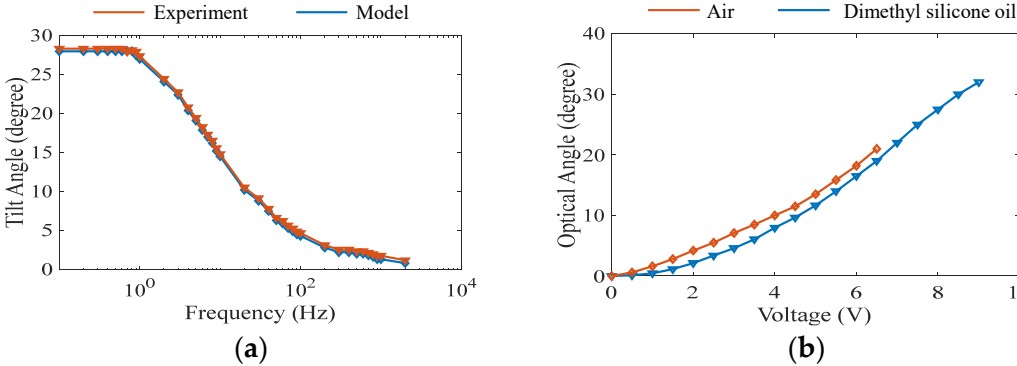

**Figure 8.** (**a**) The frequency response and (**b**) the static response in the dimethyl silicone oil.

Figure 9 shows a comparison between the predicted and the actually measured results for a step response. Due to the low cutoff frequency of the immersed micromirror, the frequencies of the driving square-wave signal of 0–6 V were chosen as 0.05 Hz, 0.5 Hz and 1 Hz, respectively. The temperature of the environment was controlled at 295 K and the pressure was 1 atm. The measured step responses are plotted in Figure 9. The response time of the immersed MEMS mirror was 400 ms, which was significantly longer than the 50 ms response time measured in the air [13] because of the large viscosity. As shown in Figure 9a,b, the predicted step response matches the measured data very well. At a higher scanning frequency, as shown in Figure 9c, the prediction error was slightly large but still less than 4%. Thus, the proposed model is effective in calculating the step response of the scanning micromirror at a wide range of frequency.

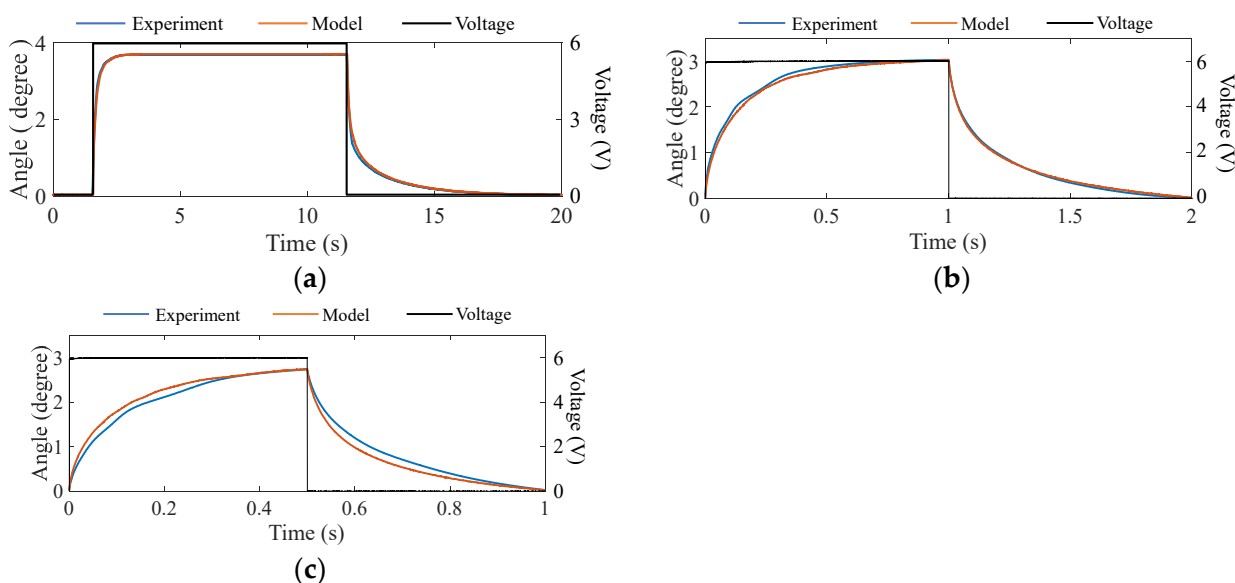

**Figure 9.** Experimental and calculated result of (**a**) 0–6 V at 0.05 Hz; (**b**) 0–6 V at 0.5 Hz; and (**c**) 0–6 V at 1 Hz.

Note that the fall time of the MEMS mirror in Figure 9a is much longer than its rise time, which is caused by the heat conduction and the non-isothermal flow. According to the damping model, the damping increases as the temperature decreases. The damping is much larger when the system returns to the zero deflection angle position at which the temperature of the bimorph actuators is lowest.

According to the above experimental results, the frequency response of the immersed MEMS mirror is largely different from that working in the air. The cutoff frequency in the liquid is far lower than that in the air and no resonance is observed. This is caused by the large viscosity in a liquid. Thus, the MEMS mirror in the liquid cannot scan with a high speed. As discussed previously, the FOV can be increased by a factor of 1.6. The immersed MEMS mirror can be used for the scenes where large FOV is needed but the scanning speed is not crucial, e.g., when the target moves slowly or is stationary.

**5. Conclusions**

In this work, a model for the damping of an electrothermally actuated MEMS mirror immersed in a liquid is established. Subsequently, the thermal and mechanical response of the MEMS mirror are obtained by incorporating the dynamic model of the MEMS mirror. Experimental results show that the proposed model is effective in calculating the step response of the scanning micromirror at a wide range of frequencies. The error between the modeling and experiment results is less than 4%. The FOV of the MEMS mirror is increased by a factor of 1.6. The cutoff frequency of the immersed MEMS mirror is much lower than that in air. Additionally, the response time in liquid is about 10 times longer than that in air. Considering the above advantages and disadvantages of the immersed electrothermally actuated micromirror, they are very suitable for applications in which the large FOV is needed but the scanning speed is not crucial. The damping model is applicable to nearly all electrothermal MEMS mirrors immersed in a Newtonian fluid. Even if the shape of the mirror plate or the actuators is changed, this model can still be used to describe the changing trends with different liquids. Packaging electrothermally actuated micromirrors in liquid is challenging and more research is needed on this aspect. Furthermore, the study on the impact resistance of immersed electrothermally actuated micromirrors will be explored in the near future.

**Author Contributions:** Methodology, T.L.; Project administration, H.X.; Resources, H.X.; Supervision, S.Q. and H.X.; Validation, T.P. and H.Z.; Visualization, T.L.; Writing—original draft, T.L.; Writing—review and editing, H.X. All authors have read and agreed to the published version of the manuscript.

**Funding:** This research was funded by the Foshan Science and Technology Innovation Project under Grant No. 2018IT100252.

**Data Availability Statement:** No new data were created or analyzed in this study. Data sharing is not applicable to this article.

**Conflicts of Interest:** The authors declare no conflict of interest.

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
