# Peer review of "Dynamic Response Analysis of an Immersed Electrothermally Actuated MEMS Mirror"

_actuators, doi:10.3390/act12020083_

Round 1

Reviewer 1 Report

The authors have reported the dynamic response analysis of an immersed electrothermally-actuated MEMS mirror. This analysis includes energy domains, such as electrical, thermal, mechanical and fluidic. In addition, air-damping modeling is developed. This manuscript can be improved by considering the following issues:

1.-The abstract should consider the main results of the proposed analysis. Furthermore, a conclusion sentence could be incorporated.

2.-The introduction should add more discussions on the research problem. The authors should mention the electrothermally-actuated MEMS mirror's advantages and drawbacks compared with other actuation mechanisms, such as piezoelectric and electrostatic. In addition, information on the advantages and limitations of the proposed air-damping modeling should be reported. More recent references between 2018 and 2022 should be presented.

3.- The second section must improve the description of the working principle of the MEMS mirror, including the different materials, dimensions, and electrical connections. 

4.-Which were the main assumptions of the air-damping modeling? What are the effects of the stress residual due to the fabrication process on the electromechanical behavior of the MEMS mirror?

5.-The authors should consider more discussions on the main limitations or challenges of the immersed electrothermally-actuated MEMS mirror. 

6-The figures 3 and 7 could be improved using a better format such as Matlab or equivalent software.

7.- The description of the FEM model of the MEMS mirror using COMSOL could add more information on the boundary and load conditions, mesh type, material properties, and CFD conditions.

8.- Information on the fabrication of the MEMS mirror could be incorporated.

9.- The resolution and quality of Figure 6 should be improved.

10.- The discussions of the main results must be enhanced.

11.- What is the future research work?

12.- The conclusion must be enhanced based on the above comments.

Reviewer 2 Report

Authors prepared a manuscript on an interesting and very practical topic - the modeling of MEMS mirrors immersed in liquid. Nevertheless, I noticed some issues and questions that are not fully answered.

1.       In figure 2a authors provide a mass-spring model, but it is not full and does not represent the real case. Mass is not shown, and damping is also not included. From the figure, it is not clear how many degrees of freedom the model has.

2.       The authors analyze the impact of the liquid on the damping, but the damping of the mechanical system itself is not mentioned.  These issues must be explained and clarified.

3.       It would be easier to understand the authors' approach if they could include an aschematic representation of their modeling approach.

4.       Authors should provide more details about the model verification procedure. How was obtained graph presented in 7a?

5.       In figure 8 provides step response corresponding to only one case. Is it enough just to use one point for verification? Moreover, it is not clear under what conditions it is obtained since only the voltage is specified.  If possible, authors should provide more results to prove the efficiency of their model in a broader range.

6.       Conclusions should be supported by some quantitative results.

Minor remarks:

Lines 31, 33,  degree symbol should be improved.

Line 125 – fonts should be adjusted.

Round 2

Reviewer 1 Report

The authors have improved their manuscript based on the reviewer's comments. However, the resolution of Figures 1c and 5 should be improved. In addition, the values of temperature in Figure 5 are blurry. Figure 5 should be replaced by another Figure with better quality.

Author Response

Dear Reviewer,

Thanks very much for taking your time to review this manuscript. I really appreciate all your comments and suggestions! We changed the pictures in Figures 1(c) and 5. Please find my responses in the attachment.

Sincerely,

Tailong Liu.

Reviewer 2 Report

Authors resolved all major mentioned issues. In my opinion, publication could be accepted for publishing after correcting some minor typing and formatting issues: lines 137,  210-221, 303-304.

Author Response

Dear Reviewer,

Thanks very much for taking your time to review this manuscript. I really appreciate all your comments and suggestions! We modified the  typing and formatting issues.

Sincerely,

Tailong Liu.